# Quantitative Analysis of the Trade-Offs of Colony Formation for *Trichodesmium*

Vitul Agarwal,[a] Keisuke Inomura,[a] Colleen B. Mouw[a]

aGraduate School of Oceanography, University of Rhode Island, Narragansett, Rhode Island, USA

**ABSTRACT** There is considerable debate about the benefits and trade-offs for colony formation in a major marine nitrogen fixer, *Trichodesmium*. To quantitatively analyze the trade-offs, we developed a metabolic model based on carbon fluxes to compare the performance of *Trichodesmium* colonies and free trichomes under different scenarios. Despite reported reductions in carbon fixation and nitrogen fixation rates for colonies relative to free trichomes, we found that model colonies can outperform individual cells in several cases. The formation of colonies can be advantageous when respiration rates account for a high proportion of the carbon fixation rate. Negative external influence on vital rates, such as mortality due to predation or micronutrient limitations, can also create a net benefit for colony formation relative to individual cells. In contrast, free trichomes also outcompete colonies in many scenarios, such as when respiration rates are equal for both colonies and individual cells or when there is a net positive external influence on rate processes (i.e., optimal environmental conditions regarding light and temperature or high nutrient availability). For both colonies and free trichomes, an increase in carbon fixation relative to nitrogen fixation rates would increase their relative competitiveness. These findings suggest that the formation of colonies in *Trichodesmium* might be linked to specific environmental and ecological circumstances. Our results provide a road map for empirical studies and models to evaluate the conditions under which colony formation in marine phytoplankton can be sustained in the natural environment.

**IMPORTANCE** *Trichodesmium* is a marine filamentous cyanobacterium that fixes nitrogen and is an important contributor to the global nitrogen cycle. In the natural environment, *Trichodesmium* can exist as individual cells (trichomes) or as colonies (puffs and tufts). In this paper, we try to answer a longstanding question in marine microbial ecology: how does colony formation benefit the survival of *Trichodesmium*? To answer this question, we developed a carbon flux model that utilizes existing published rates to evaluate whether and when colony formation can be sustained. Enhanced respiration rates, influential external factors such as environmental conditions and ecological interactions, and variable carbon and nitrogen fixation rates can all create scenarios for colony formation to be a viable strategy. Our results show that colony formation is an ecologically beneficial strategy under specific conditions, enabling *Trichodesmium* to be a globally significant organism.

**KEYWORDS** *Trichodesmium*, colony formation, marine phytoplankton, nitrogen fixation

Address correspondence to Vitul Agarwal, vitulagarwal@uri.edu.

The authors declare no conflict of interest.

*Trichodesmium* is a globally significant marine cyanobacterium that is prominent for its role as a diazotroph and in affecting the marine nitrogen cycle (1, 2). In the natural environment, *Trichodesmium* has a wide global distribution and is found in waters where bioavailable nitrogen sources are scarce (3, 4). Although *Trichodesmium* can account for about half of the biological nitrogen fixation in some areas (5), recent work has also demonstrated the widespread distribution and abundance of nondiazotrophic species of *Trichodesmium* (6). The broad importance of this cyanobacterium has led to decades of research on its ecology, physiology, and genetics (7).

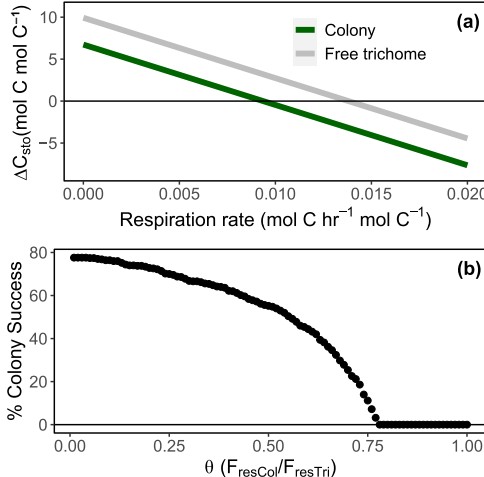

**FIG 1** (a) Change in total carbon storage ($\Delta C_{sto}$) for both colonies and free trichomes at the end of a 30-day model simulation with fixed and equal respiration costs. (b) Proportion of simulations where colonies outcompete free trichomes as $\theta$ is varied. $\theta$ is the ratio of the respiration rate of colonies and the respiration rate of free trichomes.

*Trichodesmium* is found both as free trichomes and as colonies in the natural environment (8, 9). Colonies can appear as puffs or tufts, and sizes can reach up to 2,000 $\mu$m. Although the specific reasons for colony formation are unclear, some have suggested that colonies allow *Trichodesmium* to create low-$O_2$ environments for nitrogen fixation (10). Empirical evidence, however, disputes this claim (11) and has shown that nitrogen fixation in *Trichodesmium* is lower in colonies than in free trichomes (12). The benefits of colony formation in *Trichodesmium* could be related to increased iron and phosphorus uptake (13–15), the role of microbial interactions (16), or a reduction in grazing (17). Meanwhile, the trade-offs of colony formation include lower carbon and nitrogen fixation rates (12) and $CO_2$ limitations due to lowered diffusion rates (18).

To quantitatively examine the trade-offs of colony formation, we performed modeling exercises with a carbon flux model. Quantitative models for *Trichodesmium* have been previously used to understand the coexistence of nitrogen fixation and photosynthesis (19), as well to clarify mechanisms controlling physiological processes (20) and predict the influence of growth conditions on diazotrophy (21). Conceptually simple, coarse-grained models often provide insight into ecological constraints and are a crucial link between experiments and theory (22).

With the model, we aimed to answer the following: (i) when do *Trichodesmium* colonies outcompete free trichomes? and (ii) what evolutionary pressures could drive colony formation in marine phytoplankton? To address these questions, we relied on a stepwise process that utilized published carbon fixation and nitrogen fixation rates for both colonies and free trichomes (12). We added various types of respiratory costs, an interaction term that can broadly represent effects such as grazing and mutualism, as well as alternative ratios of carbon and nitrogen fixation rates, to find scenarios where colonies outcompete trichomes in model simulations.

## RESULTS

We found that when respiration costs for both colonies and free trichomes are fixed and equal, the difference in carbon and nitrogen fixation rates favors free trichome formation (Fig. 1a). All simulations of our model result in free trichomes with better outcomes than colonies. As the respiration cost of the cell increases, the amount of C stored in the cells decreases. At the end of 30 days, respiration rates above 0.009 mol C h$^{-1}$ mol C$^{-1}$ lead to a net loss in C storage for colonies, whereas rates above 0.0135 mol C h$^{-1}$ mol C$^{-1}$ lead to a net loss in C storage for free trichomes. When respiration rates for free trichomes are fixed but those for colonies are allowed to vary proportionately, the ability of colonies to outcompete free

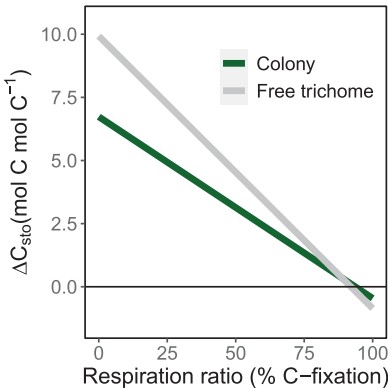

**FIG 2** Change in total carbon storage for both colonies and free trichomes at the end of a 30-day model simulation with respiration costs scaled to a proportion of the total carbon fixation rate (0.015 mol C h$^{-1}$ mol C$^{-1}$ for free trichomes and 0.010 mol C h$^{-1}$ mol C$^{-1}$ for colonies).

trichomes decreases as the respiration rates approach parity. Colony respiration rates should be lower than ~77% of the respiration rates of free trichomes to have any chance of success at the end of a 30-day simulation. This limit also represents the maximum possible success rate of colonies. In essence, it is dependent on the number of simulated opportunities for colonies to outcompete free trichomes. For models run with a higher limit of respiration rate (i.e., more simulations), the maximum possible success rate of colonies proportionately increases (see Fig. S1 in the supplemental material).

In the case where respiration rates are tied to the carbon fixation rates, higher respiration rates lead to better outcomes for colonies (Fig. 2). For rates that are greater than 88.8% of the carbon fixed in cells, colonies outperform free trichomes at the end of 30 days. There is a net loss for carbon storage for both colonies and free trichomes under high metabolic costs. Colonies can sustain losses of up to 93.6% of their carbon fixation rate and still maintain biomass, whereas free trichomes can only sustain losses of up to 92% of their carbon fixation rate.

We introduced a general parameter, $i_p$, that simulated the cumulative effects of external influence on the carbon and nitrogen fixation rates for *Trichodesmium*. In model simulations run with this additional parameter, free trichomes tend to outperform colonies when conditions allow for higher carbon and nitrogen fixation rates. Under scenarios where there would be no metabolic costs, the performance of colonies can never match that of the free trichomes (Fig. 3a). In more realistic scenarios where metabolic costs are scaled to the carbon fixation rates, the curve for colony performance moves upwards and increases the relative ability of colonies to outcompete free trichomes at the end of a 30-day simulation (Fig. 3b to d). When metabolic demands are relatively equal in magnitude (Fig. 2; 90% of their respective carbon fixation rate), we found that positive interactions favor free trichomes whereas negative interactions favor colony formation (Fig. 3d). The combination of high metabolic demand and negative external influence tends to result in a deficit for carbon storage for both colonies and free trichomes.

In the scenario where external influence, $i_t$, affects the total biomass but not the metabolic rates, free trichomes outperform colonies when respiration rates account for up to 60% of the carbon fixation rate (Fig. 4). This competitive advantage exists for both positive and negative interactions. Only in the case where respiration is 90% of the carbon fixation rate do colonies outcompete free trichomes. Colonies perform better with both positive and negative external influence on total biomass when respiration rates are relatively high.

We also created a parameter, $\tau$, that was the ratio of C fixation rates to N fixation rates. For this scenario, different relative ratios of $\tau$ were used to test whether colonies outperform free trichomes in the absence of any external factors. Colony performance decreases exponentially as the relative ratio of C and N fixation increases (Fig. 5). When evaluated for all possibilities of metabolic costs, the performance of colonies does not exceed ~42% of simulations, despite low values of $\tau_{trichome}/\tau_{colony}$. Assuming that metabolic costs are high, colonies could

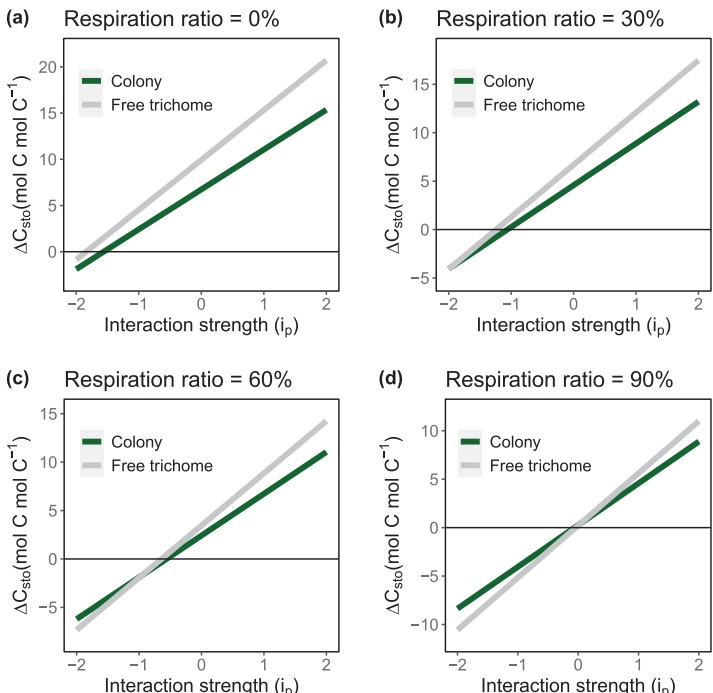

**FIG 3** Role of external influence on metabolic rates (defined as interaction strength [$i_p$]; equation 5) on carbon and nitrogen fixation rates. Each panel shows the change in total carbon storage for both colonies and free trichomes at the end of a 30-day model simulation with no metabolic costs (a), 30% of the C fixation consumed (b), 60% of the C fixation consumed (c), and 90% of the C fixation consumed (d).

theoretically improve their chances by adopting one of two strategies: increasing C fixation rates ($\tau_{colony}\uparrow$) or decreasing N fixation rates ($\tau_{colony}\uparrow$). Alternatively, colony performance is also better when trichomes decrease C fixation rates ($\tau_{trichome}\downarrow$) or increase N fixation rates ($\tau_{trichome}\downarrow$). For each of these responses, there is a maximum limit beyond which colonies always get outcompeted by free trichomes. Values of $\tau_{trichome}/\tau_{colony}$ cannot exceed 1.5 for any possibility that the colonies outcompete free trichomes. The values of $\tau$ from empirical C fixation and N fixation rates placed the relative ratio at 0.8, where our model suggests that colonies would succeed against free trichomes up to ~14% of the time.

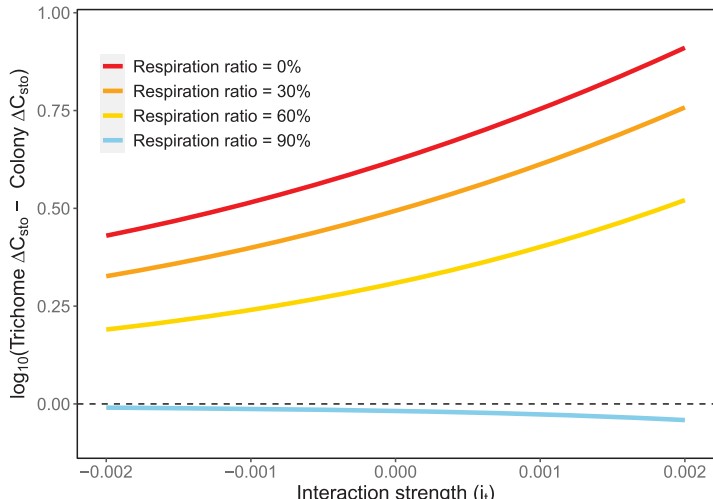

**FIG 4** Role of external influence on total biomass (defined as interaction strength [$i_t$]; equation 7) in carbon and nitrogen fixation rates. Each line shows the $\log_{10}$ difference in total carbon storage (mol C mol C$^{-1}$) for both colonies and free trichomes at the end of a 30-day model simulation. Models were run with no metabolic costs (red), 30% of the C fixation consumed (orange), 60% of the C fixation consumed (gold), or 90% of the C fixation consumed (sky blue).

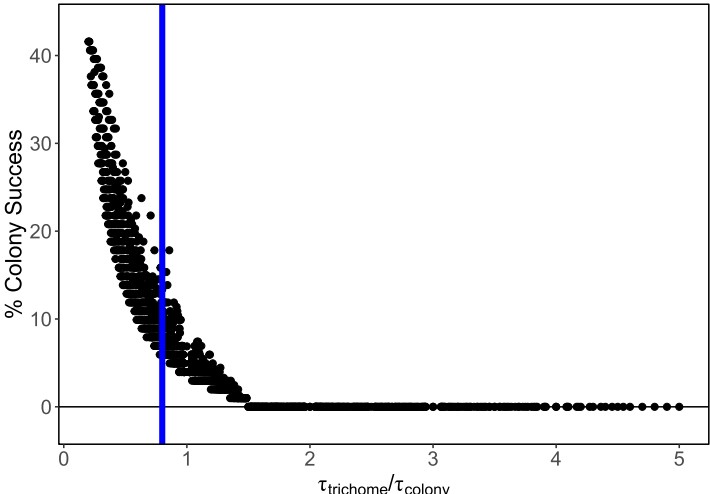

**FIG 5** Effect of variable C and N fixation ratios ($\tau$) on colony success under all possible respiration rates. Black circles are the proportion of simulations where the carbon storage of colonies was greater than that of free trichomes for a specific combination of $\tau_{colony}$ and $\tau_{trichome}$. The blue line was calculated from the reported rates (12). All possible combinations for $\tau_{trichome}$ and $\tau_{colony}$ were tested with respiration costs from 0% to 100% of the C fixation.

## DISCUSSION

**Respiration rates.** On average, *Trichodesmium* colonies have lower carbon and nitrogen fixation rates than free trichomes (12). With all else being equal, this implies that *Trichodesmium* colonies are always outcompeted by free trichomes. Figure 1a highlights this possibility for a range of fixed respiration rates. Respiration rates of >0.009 mol C h$^{-1}$ mol C$^{-1}$ lead to a net loss in carbon storage for colonies over a period of 30 days, and similarly, rates of >00.0135 mol C h$^{-1}$ mol C$^{-1}$ lead to a net loss of carbon for free trichomes. This means that colonies hit net negative in total carbon storage before free trichomes. In such cases where respiration rates are invariant, *Trichodesmium* colonies can only outcompete free trichomes in some circumstances: respiration rates of colonies must be significantly lower than that of free trichomes.

As respiration rates in phytoplankton can vary with time and space (23, 24), both *Trichodesmium* colonies and free trichomes can have different respiration rates driven by physiological or ecological factors. The respiration rate is likely to be affected by the level of oxygen management because respiratory protection (extra respiration to decrease intracellular oxygen) has been predicted to account for a large part of carbon loss (19, 25). Regardless of the contributing factor, we attempted to quantify the decrease needed for colonies to compete with free trichomes. In the absence of other variables that might influence carbon fluxes, respiration rates in colonies must be a fraction of the rates in free trichomes. Interestingly, higher possible respiration rates for *Trichodesmium* increase the maximum limit of colonies to metabolically compete with free trichomes. Our model, although coarse in resolution, suggests that *Trichodesmium* colonies have better opportunities under conditions that maximize carbon loss, regardless of the mechanism involved.

We also chose to test the scenario where metabolic costs were relative to the carbon fixation rate of both colonies and free trichomes. In this scenario, both *Trichodesmium* colonies and free trichomes can modulate their respiration rates based on the available carbon that is fixed. Colonies performed better than free trichomes for respiration rates that are greater than 88.8% of carbon fixation rates (Fig. 2). Colony formation might be a successful strategy when environmental or ecological circumstances lead to high metabolic costs. In marine phytoplankton, respiration becomes an increasingly higher proportion of carbon fixation as temperatures are increased (26). When respiration costs change proportionately with carbon fixation rates, colony formation in *Trichodesmium* might be a mechanism of reducing net carbon loss over longer periods of time. This would be because lower carbon fixation rates also lead to lower respiration rates, which minimize the loss of stored carbon (Fig. 2). Our results suggest that colony formation might have evolved in environments that are prone to

periods of high metabolic stress. In such situations, lowering carbon fixation rates, and by association, respiration rates, would allow *Trichodesmium* colonies to persist for longer than free trichomes.

**External drivers.** Colony formation in marine phytoplankton can be subject to various environmental and ecological drivers, such as temperature constraints, turbulence, predator cues, or nutrient availability (27–29). Colony adaptations can often be species-specific, with additive effects when phytoplankton are exposed to more than one environmental change (30). We added a parameter to our model to test the influence of external drivers on colony performance in *Trichodesmium*. By incorporating a general "interaction" term, our goal was to quantify the rates at which an increase or decrease in net carbon and nitrogen fixation could potentially provide advantages in forming colonies over free trichomes.

In general, we found that a large negative effect on carbon and nitrogen fixation tends to favor colonies (Fig. 3). Negative influence on carbon and nitrogen fixation rates can stand for environmental stress or micronutrient limitations. Similar to the case with varying respiration rates, the lower starting carbon and nitrogen fixation rates of colonies allow for colonies to outperform free trichomes by the end of model simulations. Cultures of *Trichodesmium* typically produce colonies after the exponential growth phase and under nutrient limitations (31, 32). By lowering their investment in carbon and nitrogen fixation rates, colonies could be employing a trade-off that increases their survival under some conditions at the cost of slower growth.

Conversely, conditions that would lead to an increase in carbon and nitrogen fixation rates appear to favor free trichomes (Fig. 3; $i_p > 0$). A positive interaction strength could stand for the ready availability of nutrients or optimal light and temperature conditions. Such conditions could allow for quicker increases in biomass for free trichomes relative to colonies. Interestingly, the relative advantage of free trichomes is significantly reduced in simulations where respiration rates are high relative to the carbon fixation rates (Fig. 3d). In this scenario, the increase in carbon fixation rate is muted by a similar, proportionate increase in metabolic costs. This means that if free trichomes were to dominate in an environment, any increases in carbon and nitrogen fixation would have to happen with minimal metabolic costs of enacting those increases. For example, an increase in nitrogen fixation rates due to ocean warming (33) might not alter the relative success of free trichomes, as thermal compensation would also increase the metabolic costs of the cells (26). On the other hand, elevated $CO_2$ concentrations have also been shown to increase nitrogen fixation rates in *Trichodesmium* (34), and it is possible that such increases benefit the formation of free trichomes over colonies. Earth system models would need to resolve potential changes in *Trichodesmium* trichome and colony abundance (and, by association, bulk C fixation and N fixation) to reduce uncertainty and improve predictions (35–37).

*Trichodesmium* has an associated epibiont of microbes that significantly extends its metabolic potential (38). Some studies have shown that *Trichodesmium* colonies can interact with bacteria to preferentially acquire iron and phosphorus from dust particles (39, 40). As there exists significant diversity in the *Trichodesmium* epibiont (41), there might be cases where different epibiont communities in colonies and free trichomes lead to mixed responses to the same environmental change.

In model simulations in which external influence directly affects the total carbon stored ($i_t$) and not the metabolic rates, our models still showed free trichomes outperforming colonies in all cases in which respiration rates are comparatively low (Fig. 4). For high respiration rates (90% of C fixation), colonies outperformed free trichomes in all cases, regardless of the direction of influence. As environmental influence in the natural environment can be a chaotic and dynamic process, there probably exist environmental drivers that affect both metabolic rates and total carbon. Our results suggest that even under cases of high carbon loss due to mortality, respiration rates are a larger factor contributing to the success of colony formation in *Trichodesmium*.

Colony formation in marine phytoplankton provides benefit against grazers (42), and it is possible that *Trichodesmium* colony formation can reduce the rates of predation by particular classes of zooplankton, possibly due to biotoxin accumulation (43, 44). The reduction of

grazing could boost colony success against free trichomes, but such a reduction should compensate for the lowered C fixation and N fixation rates. Future experimental studies could test this hypothesis by measuring the growth rates of *Trichodesmium* trichomes and colonies in the presence and absence of predator cues.

**Relative metabolic rates.** Phytoplankton can change their rates of carbon and nitrogen fixation, and global change stressors, such as changes in temperature or nutrient availability, are expected to alter these rates (45). *Trichodesmium* has been shown to alter its metabolic rates in response to various stressors: high light (46), elevated carbon dioxide concentrations (34), and anthropogenic nutrient pollution (47). C and N fixation rates in *Trichodesmium* can also vary over the course of a single day (48). We tested the effects of different C and N fixation ratios ($\tau$) on the competitiveness of colonies and free trichomes to broadly evaluate their performance in a dynamic environment.

Colony formation reduces the ability of the aggregate to fix nitrogen more than it reduces their ability to fix carbon (12). This means that colonies have higher $\tau$ ratios than free trichomes, and a further increase would continue to benefit colonies in their competitiveness against free trichomes. In contrast, circumstances that increase the $\tau$ of free trichomes exponentially decrease colony success (Fig. 5, moving right on the $x$ axis). Interestingly, if the $\tau$ of free trichomes reaches 1.5 times the $\tau$ of colonies, colonies get outcompeted, given that everything else is unchanged. However, other physiological, ecological, or environmental benefits of colony formation could be oriented toward escaping competitive exclusion from free trichomes.

**Conclusions.** Colony formation in marine phytoplankton can be subject to various ecological and environmental pressures. Because *Trichodesmium* can persist as both colonies and free trichomes, identifying the trade-offs of colony formation is imperative in understanding the ecological role of *Trichodesmium* in a changing world. We simulated multiple scenarios of metabolic demand, external influence, and relative carbon and nitrogen fixation rates to compare the performance of colonies and free trichomes in a model system.

In the case where respiration rates are equal and unchanging, colonies always get outcompeted by free trichomes. When respiration rates are unequal and variable, colonies should necessarily have only a fraction of the metabolic costs of free trichomes to stand a chance of competing. If respiration rates are proportional to carbon fixation rates, colonies perform better and outcompete free trichomes when costs exceed 88.8% of the carbon fixed in the cell. Negative external influence on metabolic rates, such as micronutrient limitations, allows colonies to perform better than free trichomes, whereas positive external influence, such as mutualistic interactions or optimal environmental conditions, would benefit free trichomes more than colonies. For both colonies and free trichomes, increasing $\tau$ would increase their competitiveness in the environment; however, if the $\tau$ of free trichomes is greater than 1.5 times the $\tau$ of colonies, there are no scenarios where colonies can compete with free trichomes.

Despite lower carbon and nitrogen fixation rates, *Trichodesmium* colonies can outperform free trichomes under many circumstances. Our results lay out a framework and many testable hypotheses that could motivate further study into the ecological and environmental drivers of colony formation in marine phytoplankton. As global environmental change is expected to alter the marine environment, changes in the ability of phytoplankton taxa to form and subsist as colonies could have globally significant ecological and biogeochemical effects.

## MATERIALS AND METHODS

We developed a model to test how differences in carbon fixation and nitrogen fixation rates affected the overall performance of *Trichodesmium* colonies and free trichomes. The model evaluated total carbon stored after gains due to carbon and nitrogen assimilation and losses due to respiration. In the general case, the model equation was simply a function of net carbon flux through time.

$$\frac{dC_{\text{sto}}}{dt} = F_{\text{cfix}} - F_{\text{nfix}} Y^{\text{C:N}} R_{\text{N:C}} - F_{\text{res}} \tag{1}$$

where $C_{\text{sto}}$ is the biomass C normalized carbon storage (mol N h$^{-1}$ mol N$^{-1}$), $F_{\text{cfix}}$ is the carbon fixation rate (mol C h$^{-1}$ mol C$^{-1}$), $F_{\text{nfix}}$ is the nitrogen fixation rate (moles N per hour per mole N), $Y^{\text{C:N}}$ is the nitrogen-to-carbon conversion for N fixation (mol C mol N$^{-1}$), $R_{\text{N:C}}$ is the cellular ratio of N to C for *Trichodesmium* (mol N mol C$^{-1}$), and $F_{\text{res}}$ is the respiration rate (mol C h$^{-1}$ mol C$^{-1}$).

**TABLE 1** Reported values of metabolic rates for *Trichodesmium* (12)

| Organism form | C fixation rate (mol C h$^{-1}$ mol C$^{-1}$) | N fixation rate (mol N h$^{-1}$ mol N$^{-1}$) |
|---|---|---|
| Free trichomes | 0.015 | 0.0075 |
| Colonies | 0.010 | 0.004 |

We assumed that the system starts with 100 mol C for both colonies and free trichomes and that $R_{N:C}$ is 0.159 (49). The C cost of N fixation was based on the balance of electron requirements and the electron supply of a typical carbohydrate (e.g., glucose) (50). Consequently, we used a $Y^{C:N}$ value of 1 (51, 52). We ran the model for 30 days with hourly time steps. At the end of 30 days, we checked to see what the net change in $C_{sto}$ was. Here, we wanted to know if the colonies had net greater carbon storage at the end of 30 days. For every model run with a unique set of parameters, we defined "colony success" as the proportion of simulations where $C_{sto}$ was higher for colonies than for free trichomes at the end of 30 days.

**Fixed and equal respiration rates.** In the simplest case, we used reported carbon fixation and nitrogen fixation rates (12) to check the relative performance of colonies and free trichomes. We assumed that $F_{res}$ was the same for both and did not vary with time. Table 1 indicates the rates used for this scenario. $F_{res}$ was set to range from 0 to 0.02 mol C h$^{-1}$ mol C$^{-1}$ with increments of 0.0005 mol C h$^{-1}$ mol C$^{-1}$.

**Fixed and unequal respiration rates.** For the second case, we tested the relative performance of colonies and free trichomes if colonies had only a fraction of the respiration rate of free trichomes. Five hundred respiration rates for free trichomes were selected from a uniform distribution with limits of 0 and 0.02 mol C h$^{-1}$ mol C$^{-1}$. We then introduced a parameter, $\theta$, that controls the effective respiration rate of colonies. $\theta$ represents the relationship between the $F_{res}$ of colonies and the $F_{res}$ of free trichomes (equation 2). This ratio was varied between 0.01 and 1 with increments of 0.01.

$$\theta = \frac{F_{res\text{-colonies}}}{F_{res\text{-trichomes}}} \tag{2}$$

Outcomes were reported as the percentage of simulations that allowed for colony success (greater net carbon storage) at the end of 30 days for any specific $\theta$.

**Respiration rates vary with carbon fixation.** As respiration rates can vary for populations, we created a parameter called the respiration coefficient ($r_p$) that determines how much energy is consumed to maintain the metabolism. We assumed that

$$F_{res} = r_p \times F_{cfix} \tag{3}$$

Here, the respiration rate of colonies and free trichomes is a fraction of the carbon fixation rate. We varied the costs of respiration from 0.1% to 100% of the total carbon fixation rate for both colonies and free trichomes.

**Simulating the role of external influence on metabolic rates.** Natural phytoplankton populations are subject to ecological and environmental pressures that can affect their rates of carbon and nitrogen fixation. We introduced an additional parameter ($F_{int}$) to the model. $F_{int}$ represents the term for external influence on metabolic rates. We set $F_{int}$ proportionally to the carbon fixation and nitrogen fixation rates. The proportion is represented by $i_p$. The general term for external influence ($F_{int}$), in essence, is meant as a multiplier of the existing carbon and nitrogen fixation rates. Theoretically, this means that interactions either remove some biomass or allow for the creation of additional biomass. Positive values of $F_{int}$ indicate that the net outcome of all ecological and environmental effects led to an increase in metabolic rates, whereas negative values of $F_{int}$ indicate that the net outcome of all ecological and environmental effects led to a decrease in metabolic rates.

$$\frac{dC_{sto}}{dt} = F_{cfix} - F_{nfix} Y^{C:N} R_{N:C} - F_{res} + F_{int} \tag{4}$$

$$F_{int} = i_p \left( F_{cfix} - F_{nfix} Y^{C:N} R_{N:C} \right) \tag{5}$$

We simulated $i_p$ to range from $-200\%$ to 200% of the combined carbon fixation and nitrogen fixation rates for both colonies and free trichomes. Carbon and nitrogen fixation rates for this scenario were the same as reported in Table 1.

**Simulating the role of external influence on total biomass.** To account for the cases where external influence affects the total carbon stored (i.e., population size or biomass) instead of the metabolic rates, we set $F_{int}$ to add or remove carbon from the storage pool. $i_t$ was set to range from $-0.002$ to 0.002 h$^{-1}$.

$$\frac{dC_{sto}}{dt} = F_{cfix} - F_{nfix} Y^{C:N} R_{N:C} - F_{res} + F_{int} \tag{6}$$

$$F_{int} = i_t \times C_{sto} \tag{7}$$

**Changing relative metabolic rates.** We tested the sensitivity of colony performance to the reported rates in Table 1. For both colonies and free trichomes, we held the carbon fixation rate constant and modulated the nitrogen fixation rate to vary based on a ratio ($\tau$). $\tau$ ranged from 1 to 5, signifying environments where nitrogen fixation rates could match rates of carbon fixation, as well as environments where nitrogen fixation was

significantly reduced relative to carbon fixation rates. This scenario was simulated in the absence of external influence on either the metabolic rates or the total carbon stored.

$$\frac{F_{cfix}}{F_{nfix}} = \tau \tag{8}$$

A higher $\tau$ implies that relative N fixation rates are lower, whereas a $\tau$ of 1 implies that N fixation rates are equal to C fixation rates. We performed this analysis for all cases of respiration rates, ranging from 0% to 100% of the carbon fixation rate. Outcomes were reported as a percentage of simulations that allowed for colony success (greater net carbon storage) at the end of 30 days. For the reported rates in Table 1, $\tau$ is 2 for free trichomes and 2.5 for colonies.

**Software.** All the analyses were conducted in R version 4.1.2 (53). The "tidyverse" (54) and "cowplot" (55) packages were used.

**Code availability statement.** The code required for the analysis is available at https://github.com/vitul-agarwal-1/trichodesmium-colony-formation (https://www.doi.org/10.5281/zenodo.6558209).

## SUPPLEMENTAL MATERIAL

Supplemental material is available online only.

**SUPPLEMENTAL FILE 1**, PDF file, 0.1 MB.

## ACKNOWLEDGMENTS

This study was supported by the US National Science Foundation (OCE-2048373, subaward SUB0000525 from Princeton University to K.I.) and Rhode Island Science and Technology Advisory Council (AWD10732 to K.I.). We thank these foundations for their support.

All authors helped prepare the manuscript and approved the final version.

We declare no competing interests.

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
