## [Reviewer comments · Microbiology Spectrum]

Microbiology Spectrum

Quantitative analysis of the trade-offs of colony formation for *Trichodesmium*

Vitul Agarwal, Keisuke Inomura, and Colleen Mouw

Corresponding Author(s): Vitul Agarwal, University of Rhode Island

Review Timeline:

Submission Date:	May 31, 2022
Editorial Decision:	August 2, 2022
Revision Received:	August 25, 2022
Editorial Decision:	September 30, 2022
Revision Received:	October 4, 2022
Accepted:	October 20, 2022

Editor: Allison Veach

Reviewer(s): The reviewers have opted to remain anonymous.

Transaction Report:

DOI: <https://doi.org/10.1128/spectrum.02025-22>

August 2, 2022

Mx. Vitul Agarwal
University of Rhode Island
Graduate School of Oceanography
Narragansett, Rhode Island 02882

Re: Spectrum02025-22 (Quantitative analysis of the trade-offs of colony formation for *Trichodesmium*)

Dear Mx. Vitul Agarwal:

Thank you for submitting your manuscript to Microbiology Spectrum. Two reviewers have evaluated your work and have provided significant feedback to improve the manuscript. Specifically, I expect the authors to take special note of Reviewer 1's comments related to model components. Please consider this similar to "Major Revisions" decision. If the authors have reasoning not to move forward with a revision the reviewers queried, please provide such reasoning and substantiate it in your point-by-point response, as indicated below.

Link Not Available

Sincerely,

Allison Veach

Journals Department
Reviewer comments:

Reviewer #1 (Comments for the Author):

Summary: Agarwal and colleagues developed a coarse grained model of carbon fluxes for individual trichomes and colonial *Trichodesmium* cells to constrain the physiological and environmental interactions which give rise to colonial phenotypes. Why *Trichodesmium* cells differentiate under some conditions remains an open question, plagued by conflicting observations and experimental results.

Parameters of the model include carbon and nitrogen fixation rates and carbon losses in the form of respiratory demands. A

term was also included as a catch-all for environmental and ecological interactions, which acts proportionately on the sum of carbon and nitrogen fixation rates. Carbon and nitrogen fixation rates for trichomes and colonies were either parameterized using published data or varied according to a tunable ratio. The authors then devised a number of simulations varying the respiratory demands and interaction effects in a combinatorial fashion. Simulations of different parameterizations of the model were used to infer the conditions which favor a colonial strategy over an individual strategy, characterized by high respiratory demand in combination with strong negative interactions. An increase in the ratio of carbon fixation to nitrogen fixation in colonies relative to trichomes was also identified as a condition which benefits colony formation.

The article was generally well written, the figures were effective and high quality, and the interpretations of simulation results were clear. I do, however, have several thoughts about how the model might be improved in a revised submission. I understand that there is a clear rationale for keeping things as simple as possible in these sorts of exploratory models and, inevitably, a reviewer will ask to resolve a missing process while another will ask to remove that process. That said, I do believe in this case it may be worthwhile exploring some additional trade-offs associated with individual and colonial phenotypes.

1. Respiratory costs. Respiration provides reductant to catalyze non-spontaneous reactions, including the especially high energy demand reaction catalyzed by nitrogenase. However, in the current formulation, respiratory costs are implemented as a scalar on the carbon fixation rate. There is no basal maintenance (a rate-independent carbon loss term), and there is no respiratory cost of nitrogen fixation. These are fundamental misrepresentations of cellular metabolism, and likely important to how fitness differs between individual and colonial phenotypes. The Pirt equation might be a better place to start.
2. The colony "handicap." Carbon and nitrogen fixation rates were assigned by the values in Table 1 for many of the simulations. Because the rates for the colony are lower than the rates for individuals, it only makes sense that the colony could outcompete the individual if the loss terms were relatively low (L177-178). To me this is almost a trivial result that points to other factors not resolved by the model.
3. Carbon equivalents of nitrogen fixation. Equation 1 defines carbon specific growth rates as the sum of carbon fixation rates, carbon equivalents of nitrogen fixation rates, and carbon loss. Carbon fixation and loss are intuitive components of net changes in carbon quotas, but less so in the case of nitrogen fixation when considered only in carbon equivalents. For instance, what does the carbon yield of nitrogen fixation mean (physiologically)? Under this formulation, a cell might accumulate carbon with zero carbon fixation, which is not a meaningful result.
4. The external factor (F_{int}) is a handy catch-all term for representing a variety of positive and negative effects on fitness. F_{int} was implemented as a factor on the sum of carbon and nitrogen fixation rates, however, some environmental and ecological effects act on the quota rather than on metabolic rates. For example, one of the key factors influencing colony fitness is predator avoidance, as you state clearly in the text. Since a higher trophic level is not represented in the model, perhaps another type of mortality without closure could be used? A density dependent quadratic mortality, for example.
5. Oxygen management is crucial for *Trichodesmium*, but is not considered here. Membranes with gas selective permeability are part of the story, but microenvironments within puffs and tufts are likely also important (although this has been challenged by Eichner and colleagues and was largely written off here). Certainly, respiration is modulated to not only supply ATP for nitrogenase but also to drive local oxygen concentrations below inhibitory levels near the active site of nitrogenase complexes. Colonies lower oxygen diffusive fluxes, thereby lowering the respiratory demand of nitrogen fixation. I was surprised that oxygen was not considered in the trade-offs of individual versus colony strategies.

Minor comments:

- In general, I found the equations difficult to read. Of course this will not be an issue in the print, but please in the future check that your equations are legible for reviewers.
- Figure 2 and elsewhere- the respiration quotient might be a bit of a confusing term for some, as it more commonly refers to the stoichiometric ratio of oxygen to carbon associated with respiration, rather than its intended meaning in your context.
- Figure 3 and L147-156: The position of the experimental tau ratios is a compelling argument for the sensitivity of the relative fitness of colonies versus individuals, so I would have liked for that argument to be fleshed out a bit more. In the same vein, requiring tau to be so heavily weighted to favor the colony could be an indication that the model is not capturing some other relevant dynamics.

Reviewer #2 (Comments for the Author):

In this paper, the authors developed a simple model based on carbon fluxes to compare conditions favoring colony formation vs free trichomes in *Trichodesmium*. Models like this can be helpful in generating testable hypotheses for natural phenomena. The paper is well-written and straightforward, and I have a few questions/suggestions to improve the clarity and accessibility of the paper:

L19: Specify that these are all R codes

L50: "Published rates" - Please specify which species these published rates are for. The genus *Trichodesmium* consists of multiple species - Are there any known species-specific differences in terms of their colony formation? Is this model then limited to species with published carbon and nitrogen fixation rates?

L82: Are iron and phosphorous uptake rates known in *Trichodesmium*? If yes, can they be included in the model since they are relevant to colony formation?

L147-156: The explanation of the C and N fixation ratios in this section is very confusing. I did not understand how this is calculated until I read the materials and methods section. The authors should briefly explain the formula so that the rest of the section e.g. L152-154 makes sense. Standardizing the decrease/increase notation between the trichomes and the colonies will also help make the section clearer - e.g. when trichomes decrease C fixation rates vs colonies decrease C fixation rates (instead of increase C fixation rates) and when trichomes increase N fixation rates vs colonies increase N fixation rates (instead of decrease N fixation rates)

L203-204: It will be helpful to briefly summarize how different environmental and ecological drivers affect colony formation in *Trichodesmium* (or other marine phytoplankton), based on known studies.

L206: The goal of incorporating this "interaction" term should also be introduced in the results

L259: What is the significance of this 1.5 times limit? How does this translate to real-life ecological scenarios?

L286: What are some examples of testable hypotheses generated from this model? How can they be tested in future?

Staff Comments:

Preparing Revision Guidelines

Please return the manuscript within 60 days; if you cannot complete the modification within this time period, please contact me. If you do not wish to modify the manuscript and prefer to submit it to another journal, please notify me of your decision immediately so that the manuscript may be formally withdrawn from consideration by Microbiology Spectrum.

The reviewer comments are in blue, and our replies are in black. The page and line numbers we cite refer to the track changes version of the manuscript.

Reviewer #1 (Comments for the Author):

Summary: Agarwal and colleagues developed a coarse grained model of carbon fluxes for individual trichomes and colonial *Trichodesmium* cells to constrain the physiological and environmental interactions which give rise to colonial phenotypes. Why *Trichodesmium* cells differentiate under some conditions remains an open question, plagued by conflicting observations and experimental results.

Parameters of the model include carbon and nitrogen fixation rates and carbon losses in the form of respiratory demands. A term was also included as a catch-all for environmental and ecological interactions, which acts proportionately on the sum of carbon and nitrogen fixation rates. Carbon and nitrogen fixation rates for trichomes and colonies were either parameterized using published data or varied according to a tunable ratio. The authors then devised a number of simulations varying the respiratory demands and interaction effects in a combinatorial fashion. Simulations of different parameterizations of the model were used to infer the conditions which favor a colonial strategy over an individual strategy, characterized by high respiratory demand in combination with strong negative interactions. An increase in the ratio of carbon fixation to nitrogen fixation in colonies relative to trichomes was also identified as a condition which benefits colony formation.

The article was generally well written, the figures were effective and high quality, and the interpretations of simulation results were clear. I do, however, have several thoughts about how the model might be improved in a revised submission. I understand that there is a clear rationale for keeping things as simple as possible in these sorts of exploratory models and, inevitably, a reviewer will ask to resolve a missing process while another will ask to remove that process. That said, I do believe in this case it may be worthwhile exploring some additional trade-offs associated with individual and colonial phenotypes.

We thank you for your time reading our manuscript. We found most of the comments constructive and we have revised our manuscript accordingly. Especially, we appreciate your understanding that there is a benefit in keeping the model simple because in this way, our model can be analytically tractable, and any parameters can be informed by empirical data. Please, see our point-by-point responses to your suggestions below. Our major revisions include:

1. Additional analysis and a new figure (Figure 1B in the revised version) that explores rate-independent carbon loss due to respiration, and the trade-offs in colony formation associated with these rates.
2. Further discussion on the role of respiration in competitiveness between colonies and free trichomes.

3. An extension of our model to consider external influence on total carbon stored, instead of the metabolic rates. Here, we added a new figure (Figure 4 in the revised version) and appropriate sections to the Methods, Results and Discussion.

We also incorporate the minor comments into the revised manuscript. We believe that the revision based on your comments improved our manuscript significantly and hope that it will be found satisfactory.

1. Respiratory costs. Respiration provides reductant to catalyze non-spontaneous reactions, including the especially high energy demand reaction catalyzed by nitrogenase. However, in the current formulation, respiratory costs are implemented as a scalar on the carbon fixation rate. There is no basal maintenance (a rate-independent carbon loss term), and there is no respiratory cost of nitrogen fixation. These are fundamental misrepresentations of cellular metabolism, and likely important to how fitness differs between individual and colonial phenotypes. The Pirt equation might be a better place to start.

Thank you for the comment. In our current model setup, the costs of nitrogen fixation are already incorporated as a negative carbon flux ($-F_{nfix} Y^{C:N} R_{N:C}$). Except for Fig. 5, this term is constant, thus, essentially functioning as maintenance. Thus, this term together with C fixation dependent respiration provides a Pirt equation like formula (maintenance respiration + growth dependent part), the maintenance of which is informed by the empirical rate of nitrogen fixation. On a more detailed level, growth and photosynthesis are not identical, yet in general these are positively correlated; growth rates are often inferred from rates of photosynthetic ^{14}C uptake (McAllister et al. 1964; Malone 1982).

In addition to that, to isolate the effect of constant (analogous to maintenance type respiration) we have a scenario where the respiration rates are fixed and rate-independent (Figure 1A). As colonies have lower carbon and nitrogen fixation rates, they inherently have a disadvantage compared to individual trichomes. The only scenario where basal respiration rates may offer colonies an advantage is when their respiration rates are significantly lower than those of individual trichomes.

Furthermore, in the new manuscript, we have now added an additional figure describing this possibility in greater detail (Figure 1B; shown below). Essentially, we have extended this scenario to allow for different, but fixed, respiration rates for colonies and free trichomes. As these are fixed rates that are independent of both carbon and nitrogen fixation, we can infer the influence of increased or decreased carbon loss on the relative success of colonies (independent of the physiological causes for different respiration rates).

Figure 1: (A) Change in total carbon storage (mol C mol C^{-1}) for both colonies and free trichomes at the end of a 30-day model simulation with fixed and equal respiration costs (B) Proportion of simulations where colonies outcompete free trichomes as θ is varied. θ is the ratio of the respiration rate of colonies and the respiration rate of free trichomes. Green refers to colonies and grey refers to the free trichomes.

Malone, T. C. 1982. Phytoplankton photosynthesis and carbon-specific growth: Light-saturated rates in a nutrient-rich environment. *Limnol. Oceanogr.* **27**: 226–235.
doi:10.4319/lo.1982.27.2.0226

McAllister, C. D., N. Shah, and J. D. H. Strickland. 1964. Marine Phytoplankton Photosynthesis as a Function of Light Intensity: A Comparison of Methods. *J. Fish. Res. Board Canada* **21**: 159–181. doi:10.1139/f64-013

2. The colony "handicap." Carbon and nitrogen fixation rates were assigned by the values in Table 1 for many of the simulations. Because the rates for the colony are lower than the rates for

individuals, it only makes sense that the colony could outcompete the individual if the loss terms were relatively low (L177-178). To me this is almost a trivial result that points to other factors not resolved by the model.

We agree with the reviewer that this result might seem trivial. Our goal here was to introduce the model (in its most basic form), acknowledge the expected result, and progressively make the model more complex.

With the new figure we added, we now include additional text in the manuscript that highlights why these results might be interesting.

Lines 110-117: “When respiration rates for free trichomes are fixed, but those for colonies are allowed to vary proportionately, the ability of colonies to outcompete free trichomes decreases as the respiration rates approach parity. Colony respiration rates should be lower than ~77% of the respiration rates of free trichomes to have any chance of success at the end of a 30-day simulation. This limit also represents the maximum possible success rate of colonies. In essence, it suggests how much of the green line in Figure 1A is above the maximum value of respiration rate in the grey line. For models run with a higher limit of respiration rate, the maximum possible success rate of colonies proportionately increases (Figure S1).”

Lines 262-274: “As respiration rates in phytoplankton can vary with time and space (Robinson 2019; Mantikci et al. 2020), both *Trichodesmium* colonies and free trichomes can have different respiration rates driven by physiological or ecological factors. We attempted to quantify the decrease needed for colonies to compete with free trichomes. In the absence of other variables that might influence carbon fluxes, respiration rates in colonies must be a fraction of the rates in free trichomes. Interestingly, higher possible respiration rates for *Trichodesmium* increases the maximum limit of colonies to metabolically compete with free trichomes. Our model, although coarse in resolution, suggests that *Trichodesmium* colonies have better opportunities in conditions that maximize carbon loss, regardless of the mechanism involved.”

Figure S1: (A) Change in total carbon storage (mol C mol C^{-1}) for both colonies and free trichomes at the end of a 30-day model simulation with fixed and equal respiration costs (B) Proportion of simulations where colonies outcompete free trichomes as θ is varied. θ is the ratio of the respiration rate of colonies and the respiration rate of free trichomes. Green refers to colonies and grey refers to the free trichomes.

3. Carbon equivalents of nitrogen fixation. Equation 1 defines carbon specific growth rates as the sum of carbon fixation rates, carbon equivalents of nitrogen fixation rates, and carbon loss. Carbon fixation and loss are intuitive components of net changes in carbon quotas, but less so in the case of nitrogen fixation when considered only in carbon equivalents. For instance, what does the carbon yield of nitrogen fixation mean (physiologically)? Under this formulation, a cell might accumulate carbon with zero carbon fixation, which is not a meaningful result.

We are sorry that the explanation was confusing. We wish to note that $Y^{C:N}$ is defined as the nitrogen to carbon conversion rather than yield for N_2 fixation. In the new manuscript, we clarified it:

Line 428: “ $Y^{C:N}$ is the nitrogen to carbon conversion for N fixation (mol C mol N^{-1})”

Here the loss of C by nitrogen fixation is computed based on the electron balance. N₂ fixation consumes $4e^{-1}$ per N fixed ($8e^{-1}$ for N₂). On the other hand, complete oxidation of typical carbohydrate CH₂O may produce $4e^{-1}$, which leads to $Y^{C:N}$ of 1 (Inomura et al., 2017, 2019). This idea of electron balance is largely inspired by established methods in chemical reactions in microorganisms (Rittmann and McCarty 2001). We have added a sentence to convey the idea:

Lines 432-434: “The C cost of N₂ fixation was based on the balance of electron requirements and the electron supply of a typical carbohydrate (e.g. glucose) (Rittmann and McCarty 2001). Consequently, we used $Y^{C:N} = 1$ (Inomura et al. 2017, 2019).”

Because there is a minus sign for the nitrogen fixation term, the C yield of nitrogen fixation is negative due to the requirement for the electron; C is lost due to nitrogen fixation. In reality, electron extracted from water may be directly used in nitrogen fixation, but this process sacrifices the electron that would otherwise be used for C fixation, thus, mathematically our formula covers such a case. Regarding the concern, “Under this formulation, a cell might accumulate carbon with zero carbon fixation, which is not a meaningful result.”, we do not believe that this would be the case in this model because, given F_{Cfix} and F_{res} are zero, positive change in C_{Sto} would require negative nitrogen fixation (i.e., combination of ammonium oxidation and denitrification), which is not considered in this study nor has been observed in *Trichodesmium*.

4. The external factor (F_{int}) is a handy catch-all term for representing a variety of positive and negative effects on fitness. F_{int} was implemented as a factor on the sum of carbon and nitrogen fixation rates, however, some environmental and ecological effects act on the quota rather than on metabolic rates. For example, one of the key factors influencing colony fitness is predator avoidance, as you state clearly in the text. Since a higher trophic level is not represented in the model, perhaps another type of mortality without closure could be used? A density dependent quadratic mortality, for example.

This is an excellent suggestion and we have now incorporated additional analysis that allows for external influence on total biomass, instead of the metabolic rates. Appropriate sections have been added to the Methods, Results and Discussion. We also have added an additional figure to the manuscript (Figure 4; copied below).

Instead of only mortality, we decided to use a parameter i_t that adds or removes carbon from the total pool at each time step. This also allowed us to account for unknown cases where external influence increases the carbon available to the population.

Figure 4: The role of external influence on total biomass (defined as interaction strength i_p ; Equation 7) on carbon and nitrogen fixation rates. Each panel shows the difference in total carbon storage ($mol C mol C^{-1}$) for both colonies and free trichomes at the end of a 30-day model simulation with (A) no metabolic costs, (B) 30% of the C fixation is consumed, (C) 60% of the C fixation is consumed and (D) 90% of the C fixation is consumed.

Lines 195-200: “In the scenario where external influence i_t affects the total biomass, but not the metabolic rates, free trichomes outperform colonies when respiration rates account for up to 60% of the carbon fixation rate (Figure 4A-C). This competitive advantage exists for both positive and negative interactions. Only in the case where respiration is 90% of the carbon fixation rate do colonies outcompete free trichomes. Colonies perform better with both positive and negative external influence on total biomass when respiration rates are relatively high.”

Lines 347-363:

“In model simulations where external influence directly affects the total carbon stored (i_t) and not the metabolic rates, our models still showed free trichomes outperforming colonies in all cases when respiration rates are comparatively low (Figure 4A-C). For high respiration rates (90% of C fixation; Figure 4D), colonies outperformed free trichomes in all cases, regardless of the direction of influence. As environmental influence in the natural environment can be a chaotic and dynamic process, there probably exist environmental drivers that affect both metabolic rates and total carbon. Our results suggest that even under cases of high carbon loss due to mortality, respiration rates are a larger contributing factor to the success of colony formation in *Trichodesmium*.

Colony formation in marine phytoplankton provides benefit against grazers (Ryderheim et al. 2022) and it is possible that *Trichodesmium* colony formation can reduce the rates of predation by particular classes of zooplankton, possibly due to biotoxin accumulation (Guo and Tester 1994; Narayana et al. 2014). The reduction of grazing could boost colony success against free trichomes, but such a reduction should compensate for the lowered C fixation and N fixation rates. Future experimental studies could test this hypothesis by measuring the growth rates of *Trichodesmium* trichomes and colonies in the presence and absence of predator cues.”

5. Oxygen management is crucial for *Trichodesmium*, but is not considered here. Membranes with gas selective permeability are part of the story, but microenvironments within puffs and tufts are likely also important (although this has been challenged by Eichner and colleagues and was largely written off here). Certainly, respiration is modulated to not only supply ATP for nitrogenase but also to drive local oxygen concentrations below inhibitory levels near the active site of nitrogenase complexes. Colonies lower oxygen diffusive fluxes, thereby lowering the respiratory demand of nitrogen fixation. I was surprised that oxygen was not considered in the trade-offs of individual versus colony strategies.

Thank you for bringing this up. Oxygen management is indeed an important part of metabolic activity. At the same time, it is highly complex, and the debate has been ongoing (as the reviewer recognizes). Questions remain in oxygen management, especially regarding the difference between filaments and colonies. Whereas colonies may reduce oxygen diffusion, a recent study shows higher N₂ fixation in individual trichomes (Eichner et al 2019). Recent studies also show cases with increased O₂ concentrations near the center of the colony (possibly due to photosynthesis). Thus, including the detailed process of oxygen management in our model may require introducing ungrounded formulas and increase the uncertainty and explicit representation of oxygen management is outside of scope of the study.

We note that previous studies considering oxygen management (Inomura et al 2019; Luo et al 2022) do not differentiate between free trichomes and colonies, possibly due to the highly complex physics in colony systems and insufficient data availability to determine parameters.

To circumvent these issues, we took a different approach – testing various cases of respiration rates and plotting the results for each case. In other words, we investigated differences in oxygen management based on the level of respiration and spotted the cases where colonies gain ecological advantages. Thus, it answers a part of the key question “When do *Trichodesmium* colonies outcompete free trichomes?” from the viewpoint of respiration/oxygen management. Our approach implicitly includes oxygen management, because it is one of the major factors affecting the rate of respiration via respiratory protection (excess respiration to manage oxygen; as the reviewer points out). To convey this point and to clarify the potential effect of oxygen management, as the reviewer points out, we have mentioned oxygen management as one of controlling factors of respiration, in the new manuscript.

Lines 264-267: “The respiration rate is likely to be affected by the level of oxygen management because respiratory protection (extra respiration to decrease intracellular oxygen) has been predicted to account for a large part of carbon loss (Inomura et al., 2019; Luo et al 2022).”

We believe that this approach demonstrates the strength of the model as it allows many hypothetical cases where specific results may emerge. Similar approaches (testing various cases for unknown parameters) have been used in previous studies (e.g., Follows et al., 2007, Bruggeman and Kooijman 2007, Inomura et al 2018, Masuda et al., 2020).

Bruggeman J., Kooijman S.A. A biodiversity-inspired approach to aquatic ecosystem modeling *Limnol. Oceanogr.*, 52 (4), pp. 1533-1544 (2007)

Inomura K, Bragg J, Riemann L. et. al. A quantitative model of nitrogen fixation in the presence of ammonium. *PLOS ONE* 13(11): e0208282. (2018)
<https://doi.org/10.1371/journal.pone.0208282>

Masuda, T., Inomura, K., Takahata, N. et al. Heterogeneous nitrogen fixation rates confer energetic advantage and expanded ecological niche of unicellular diazotroph populations. *Commun Biol* 3, 172 (2020). <https://doi.org/10.1038/s42003-020-0894-4>

M. J. Follows, S. Dutkiewicz, S. Grant, S. W. Chisholm, Emergent biogeography of microbial communities in a model ocean. *Science* 315, 1843–1846 (2007).

Minor comments: -

In general, I found the equations difficult to read. Of course this will not be an issue in the print, but please in the future check that your equations are legible for reviewers. –

Thank you. We have made the equations larger in the revised manuscript.

Figure 2 and elsewhere- the respiration quotient might be a bit of a confusing term for some, as it more commonly refers to the stoichiometric ratio of oxygen to carbon associated with respiration, rather than its intended meaning in your context. –

We have changed this term to “respiration ratio” to avoid confusion. We have also altered the figures where “respiration quotient” was mentioned.

Figure 3 and L147-156: The position of the experimental tau ratios is a compelling argument for the sensitivity of the relative fitness of colonies versus individuals, so I would have liked for that argument to be fleshed out a bit more. In the same vein, requiring tau to be so heavily weighted to favor the colony could be an indication that the model is not capturing some other relevant dynamics.

Thank you for bringing this up. Our goal here was to test how variable C fixation and N fixation rates could alter colony success rates, independent of any external factors. We have added text to clarify our intent.

Lines 208-224: “We also created a parameter τ that was the ratio of C fixation rates to N fixation rates. For this scenario, different relative ratios of τ were used to test whether colonies outperform free trichomes in the absence of any external factors. In simulations that allow for different C and N fixation ratios (τ), C colony performance decreases exponentially as the relative ratio of C and N fixation increases (Figure 53). When evaluated for all possibilities of metabolic costs, the performance of colonies does not exceed ~42% of simulations, despite low values of $\tau_{\text{trichome}}/\tau_{\text{colony}}$. Assuming that metabolic costs are high, colonies could theoretically improve their chances by adopting one of two strategies – increasing C fixation rates ($\tau_{\text{colony}}\uparrow$) or by decreasing N fixation rates ($\tau_{\text{colony}}\uparrow$). Alternatively, colony performance is also better in four distinct scenarios – when trichomes decrease C fixation rates ($\tau_{\text{trichome}}\downarrow$), colonies decrease N fixation rates ($\tau_{\text{colony}}\uparrow$), or when trichomes increase N fixation rates ($\tau_{\text{trichome}}\downarrow$), or when colonies increase C fixation rates ($\tau_{\text{colony}}\uparrow$). For each of these responses, there is a maximum limit beyond which colonies always get outcompeted by free trichomes. Values of $\tau_{\text{trichome}}/\tau_{\text{colony}}$ cannot exceed 1.5 for any possibility that the colonies outcompete free trichomes. The values of τ from empirical C fixation and N fixation rates placed the relative ratio at 0.8, where our model suggests that colonies would succeed against free trichomes up to ~14% of the time.”

Lines 512-513: “This scenario was simulated in the absence of external influence on either the metabolic rates or the total carbon stored.”

Reviewer #2 (Comments for the Author):

In this paper, the authors developed a simple model based on carbon fixation to compare conditions favoring colony formation vs free trichomes in *Trichodesmium*. Models like this can be helpful in generating testable hypotheses for natural phenomena. The paper is well-written and straightforward, and I have a few questions/suggestions to improve the clarity and accessibility of the paper:

Thank you for the time reading our manuscript. We are encouraged by the overall positive comments. Below we include detailed response to each question. We hope that the new manuscript will be found satisfactory.

L19: Specify that these are all R codes

We added a line specifying the use of R.

Lines 525-526: “All the analyses were conducted in R version 4.1.2 (R Core Team 2021). The “tidyverse” (Wickham et al. 2019) and “cowplot” (Wilke 2020) packages were used.”

L50: "Published rates" - Please specify which species these published rates are for. The genus *Trichodesmium* consists of multiple species - Are there any known species-specific differences in terms of their colony formation? Is this model then limited to species with published carbon and nitrogen fixation rates?

The rates we refer were derived from cultures of *Trichodesmium erythraeum* IMS101. We follow the conventions of the original paper (reference below), which uses their methods to infer carbon fixation and nitrogen fixation rates for the entire genus. We hope the reviewer understands our insistence on following current convention, especially when we utilize rate data from previous studies.

Eichner, M., S. Thoms, B. Rost, W. Mohr, S. Ahmerkamp, H. Ploug, M. M. M. Kuypers, and D. de Beer. 2019. N₂ fixation in free-floating filaments of *Trichodesmium* is higher than in transiently suboxic colony microenvironments. *New Phytol.* 222: 852–863.
doi:10.1111/nph.15621

L82: Are iron and phosphorous uptake rates known in *Trichodesmium*? If yes, can they be included in the model since they are relevant to colony formation?

As these rates can depend heavily on the external environment, we prefer to keep our model simple and easy to understand. The effect of increased iron and phosphorus uptake can be inferred from the general interaction term F_{int} , which serves to increase or decrease the metabolic rates. We understand that there are studies regarding colony formation potentially facilitating Fe uptake (Qiu et al. 2022; Rubin et al. 2011). We are also aware of the study that shows relationship between colony formation and Fe and P deprivation (Tzubari et al. 2018). Yet, we believe that detailed mechanisms are still under investigation, and thus, explicit

inclusions of these processes are beyond the scope of the present study. However, our model may provide framework toward this direction.

Qiu, G. W., C. Koedooder, B. S. Qiu, Y. Shaked, and N. Keren. 2022. Iron transport in cyanobacteria – from molecules to communities. *Trends Microbiol.* **30**: 229–240. doi:10.1016/j.tim.2021.06.001

Rubin, M., I. Berman-Frank, and Y. Shaked. 2011. Dust-and mineral-iron utilization by the marine dinitrogen-fixer *Trichodesmium*. *Nat. Geosci.* **4**: 529–534. doi:10.1038/ngeo1181

Tzubari, Y., L. Magnezi, A. Be'Er, and I. Berman-Frank. 2018. Iron and phosphorus deprivation induce sociality in the marine bloom-forming cyanobacterium *Trichodesmium*. *ISME J.* **12**: 1682–1693. doi:10.1038/s41396-018-0073-5

L147-156: The explanation of the C and N fixation ratios in this section is very confusing. I did not understand how this is calculated until I read the materials and methods section. The authors should briefly explain the formula so that the rest of the section e.g. L152-154 makes sense. Standardizing the decrease/increase notation between the trichomes and the colonies will also help make the section clearer - e.g. when trichomes decrease C fixation rates vs colonies decrease C fixation rates (instead of increase C fixation rates) and when trichomes increase N fixation rates vs colonies increase N fixation rates (instead of decrease N fixation rates)

We have added text to describe τ and adopted the reviewer's suggested language.

Lines 208-224: “We also created a parameter τ that was the ratio of C fixation rates to N fixation rates. For this scenario, different relative ratios of τ were used to test whether colonies outperform free trichomes in the absence of any external factors. In simulations that allow for different C and N fixation ratios (τ), C colony performance decreases exponentially as the relative ratio of C and N fixation increases (Figure 53). When evaluated for all possibilities of metabolic costs, the performance of colonies does not exceed ~42% of simulations, despite low values of $\tau_{\text{trichome}}/\tau_{\text{colony}}$. Assuming that metabolic costs are high, colonies could theoretically improve their chances by adopting one of two strategies – increasing C fixation rates ($\tau_{\text{colony}}\uparrow$) or by decreasing N fixation rates ($\tau_{\text{colony}}\uparrow$). Alternatively, colony performance is also better in four distinct scenarios – when trichomes decrease C fixation rates ($\tau_{\text{trichome}}\downarrow$), colonies decrease N fixation rates ($\tau_{\text{colony}}\uparrow$), or when trichomes increase N fixation rates ($\tau_{\text{trichome}}\downarrow$), or when colonies increase C fixation rates ($\tau_{\text{colony}}\uparrow$). For each of these responses, there is a maximum limit beyond which colonies always get outcompeted by free trichomes. Values of $\tau_{\text{trichome}}/\tau_{\text{colony}}$ cannot exceed 1.5 for any possibility that the colonies outcompete free trichomes. The values of τ from empirical C fixation and N fixation rates placed the relative ratio at 0.8, where our model suggests that colonies would succeed against free trichomes up to ~14% of the time.”

L203-204: It will be helpful to briefly summarize how different environmental and ecological drivers affect colony formation in *Trichodesmium* (or other marine phytoplankton), based on known studies.

We have now added some more text and references to this section.

Lines 291-299: “Colony formation in marine phytoplankton can be subject to varying environmental and ecological drivers, such as temperature constraints, turbulence, predator cues or nutrient availability (Dell’aquila et al. 2017; Kenitz et al. 2020; Rigby and Selander 2021). Colony adaptations can often be species-specific, with additive effects when phytoplankton are exposed to more than one environmental change (Rigby et al. 2022).”

L206: The goal of incorporating this "interaction" term should also be introduced in the results

We added a line specifying the additional parameter in the Results section.

Lines 152-153: “We introduced a general parameter i_p that simulated the cumulative effects of external influence on the carbon and nitrogen fixation rates for *Trichodesmium*. In model simulations run with this additional parameter,”

L259: What is the significance of this 1.5 times limit? How does this translate to real-life ecological scenarios?

The 1.5 limit could be reached from several different mechanisms: (i) free trichomes could boost their C fixation rates (such as in bloom conditions), or, (ii) colonies could have reduced C fixation under stressful conditions. As our study relies on models, it is difficult to directly translate the significance of theoretical limits to real-life ecological scenarios. We have clarified that this section relies on variable metabolic rates under no external influence.

We have also mentioned the location of empirical studies in regard to this ratio and would prefer the general readership to evaluate this limit using their own experimental protocols.

Lines 208-213: “For this scenario, different relative ratios of τ were used to test whether colonies outperform free trichomes in the absence of any external factors.”

Lines 222-224: “The values of τ from empirical C fixation and N fixation rates placed the relative ratio at 0.8, where our model suggests that colonies would succeed against free trichomes up to ~14% of the time.”

L286: What are some examples of testable hypotheses generated from this model? How can they be tested in future?

Our results suggest that respiratory stress and external influence can play a significant role in determining the relative success of colonies, both of which can be extensively tested in experimental settings. We also found a natural limit in relative carbon fixation to nitrogen fixation ratios that implies colonies always get outcompeted in some circumstances. Future studies on *Trichodesmium* and other genera could ideally calculate these ratios and test the boundaries of these constraints. We have added some text discussing some testable hypotheses.

Lines 337-340: “Earth system models would need to resolve potential changes in *Trichodesmium* trichome and colony abundance (and by association, bulk C fixation and N fixation) to reduce uncertainty and improve predictions (Riche and Christian 2018; Tang et al. 2019; Wrightson and Tagliabue 2020).”

Lines 360-363: “The reduction of grazing could boost colony success against free trichomes, but such a reduction should compensate for the lowered C fixation and N fixation rates. Future experimental studies could test this hypothesis by measuring the growth rates of *Trichodesmium* trichomes and colonies in the presence and absence of predator cues.”

September 30, 2022

Mx. Vitul Agarwal
University of Rhode Island
Graduate School of Oceanography
Narragansett, Rhode Island 02882

Re: Spectrum02025-22R1 (Quantitative analysis of the trade-offs of colony formation for *Trichodesmium*)

Dear Mx. Vitul Agarwal:

Thank you for submitting your manuscript to Microbiology Spectrum. As you will see your paper is very close to acceptance. Please modify the manuscript along the lines I have recommended. As these revisions are quite minor, I expect that you should be able to turn in the revised paper in less than 30 days, if not sooner.

Editor Comments & Suggestions:

I appreciate the authors organization in their response to reviewers and in the manuscript itself. I have only minor comments for you to consider.

-Throughout the MS, the authors describe graphs not in biological terms, but more so explaining specific graphical variables. For example, Line 111 reads "In essence, it suggests how much of the green line in Figure 1A is above the maximum value of respiration rate in the grey line." I'm not sure how that relates to the information before it which state that ability of colonies to outcompete free trichomes decreases with respiration as it approaches parity. Can the authors please explain this more simply or effectively relate this to Figure 1 in simpler terms more so?

-Other issues as above are seen in the discussion as well, where reference to colored lines in figures is written out. Please consider changing to explaining the figure in biological terms.

-Line 91: This is not a big deal, but I don't think you should keep referring to this as a "simple model." It may be not as complex as the relationship between free-living and colony *Trichodesmium* metabolism occurs in the environment, but this model is useful in understanding potential outcomes for a range of conditions. Performing model simulations is an important step in work like this!

-Can you consider combining Figure 4A-C into one panel and color code the different respiration ratios so that this figure is easier to compare among differences.

When submitting the revised version of your paper, please provide (1) point-by-point responses to the issues raised by the reviewers as file type "Response to Reviewers," not in your cover letter, and (2) a PDF file that indicates the changes from the original submission (by highlighting or underlining the changes) as file type "Marked Up Manuscript - For Review Only". Please use this link to submit your revised manuscript. Detailed instructions on submitting your revised paper are below.

Link Not Available

Sincerely,

Allison Veach

Reviewer comments:

Preparing Revision Guidelines

Please return the manuscript within 60 days; if you cannot complete the modification within this time period, please contact me. If you do not wish to modify the manuscript and prefer to submit it to another journal, please notify me of your decision immediately so that the manuscript may be formally withdrawn from consideration by Microbiology Spectrum.

The reviewer comments are in blue, and our replies are in black. The page and line numbers we cite refer to the track changes version of the manuscript.

Editor Comments & Suggestions:

I appreciate the authors organization in their response to reviewers and in the manuscript itself. I have only minor comments for you to consider.

Thank you for your time reviewing the manuscript. We appreciate the additional comments, which we found constructive. We respond to each point below and have accepted all the proposed changes. We believe that the new revisions further reinforce the study with improved clarity.

-Throughout the MS, the authors describe graphs not in biological terms, but more so explaining specific graphical variables. For example, Line 111 reads "In essence, it suggests how much of the green line in Figure 1A is above the maximum value of respiration rate in the grey line." I'm not sure how that relates to the information before it which state that ability of colonies to outcompete free trichomes decreases with respiration as it approaches parity. Can the authors please explain this more simply or effectively relate this to Figure 1 in simpler terms more so?

I am sorry for the confusion. Following the reviewer's suggestion, we have now rephrased this section. We are trying to show that the success limit for colonies depends on the maximum possible respiration rate.

Ln 114-117: "In essence, it is dependent on the number of simulated opportunities for colonies to outcompete free trichomes. For models run with a higher limit of respiration rate (i.e. more simulations), the maximum possible success rate of colonies proportionately increases (Figure S1)."

-Other issues as above are seen in the discussion as well, where reference to colored lines in figures is written out. Please consider changing to explaining the figure in biological terms.

We thank the reviewer's suggestion. We have edited the discussion to remove references to colored lines. When required, appropriate biological terms were used.

Ln 233: "Figure 1A highlights this possibility for a range of fixed respiration rates."

-Line 91: This is not a big deal, but I don't think you should keep referring to this as a "simple model." It may be not as complex as the relationship between free-living and colony Trichodesmium metabolism occurs in the environment, but this model is useful in understanding potential outcomes for a range of conditions. Performing model simulations is an important step in work like this!

We appreciate your support. On re-reading the manuscript, we also realize that we may have overused the term “simple model”. We now reduce our usage of “simple” model by removing the adjective.

Ln 24-26: “To quantitatively analyze the trade-offs, we developed a metabolic model based on carbon fluxes to compare the performance of *Trichodesmium* colonies and free trichomes under different scenarios.”

Ln 50-51: “To answer this question, we developed a carbon flux model that utilizes existing published rates to evaluate whether and when colony formation can be sustained.”

Ln 87: “To quantitatively examine the trade-offs of colony formation, we performed modeling exercises with a carbon flux model”

Ln 94: “With the model, we aimed to answer [1] When do *Trichodesmium* colonies outcompete free trichomes? and [2] What evolutionary pressures could drive colony formation in marine phytoplankton?”

Ln 369: “We developed a model to test how differences in carbon fixation and nitrogen fixation rates affected the overall performance of *Trichodesmium* colonies and free trichomes.”

Ln 372: “In the general case, the model equation was simply a function of net carbon flux through time.”

-Can you consider combining Figure 4A-C into one panel and color code the different respiration ratios so that this figure is easier to compare among differences.

Thank you for this great suggestion. Attached below is a copy of the new figure 4 where all the lines are color-coded on the same panel. We also modified the y-axis to a log₁₀-scale to ease the comparison.

Figure 4: The role of external influence on total biomass (defined as interaction strength i_t ; Equation 7) on carbon and nitrogen fixation rates. Each line shows the \log_{10} difference in total carbon storage (mol C mol C^{-1}) for both colonies and free trichomes at the end of a 30-day model simulation. Models were run with no metabolic costs (red), with 30% of the C fixation consumed (orange), 60% of the C fixation consumed (gold) or, with 90% of the C fixation is consumed (sky blue).

October 20, 2022

Mx. Vitul Agarwal
University of Rhode Island
Graduate School of Oceanography
Narragansett, Rhode Island 02882

Re: Spectrum02025-22R2 (Quantitative analysis of the trade-offs of colony formation for *Trichodesmium*)

Dear Mx. Vitul Agarwal:

Thank you for your revisions! Your manuscript has been accepted, and I am forwarding it to the ASM Journals Department for publication. You will be notified when your proofs are ready to be viewed.

Sincerely,

Allison Veach
Editor, Microbiology Spectrum
